# Germination at Extreme Temperatures: Implications for Alpine Shrub Encroachment

**DOI:** 10.3390/plants10020327

**Published:** 2021-02-09

**Authors:** Susanna E. Venn, Rachael V. Gallagher, Adrienne B. Nicotra

**Affiliations:** 1Centre for Integrative Ecology, School of Life and Environmental Sciences, Deakin University, Burwood, VIC 3125, Australia; 2Research School of Biology, Australian National University, Acton, ACT 2600, Australia; adrienne.nicotra@anu.edu.au; 3Department of Biological Sciences, Macquarie University, North Ryde, NSW 2109, Australia; rachael.gallagher@mq.edu.au

**Keywords:** germination niche, temperature gradient plate, climate extremes, conservation management, species geographic range, climate warming, Australia

## Abstract

Worldwide, shrub cover is increasing across alpine and tundra landscapes in response to warming ambient temperatures and declines in snowpack. With a changing climate, shrub encroachment may rely on recruitment from seed occurring outside of the optimum temperature range. We used a temperature gradient plate in order to determine the germination niche of 14 alpine shrub species. We then related the range in laboratory germination temperatures of each species to long-term average temperature conditions at: (1) the location of the seed accession site and (2) across each species geographic distribution. Seven of the species failed to germinate sufficiently to be included in the analyses. For the other species, the germination niche was broad, spanning a range in temperatures of up to 17 °C, despite very low germination rates in some species. Temperatures associated with the highest germination percentages were all above the range of temperatures present at each specific seed accession site. Optimum germination temperatures were consistently within or higher than the range of maximum temperatures modelled across the species’ geographic distribution. Our results indicate that while some shrub species germinate well at high temperatures, others are apparently constrained by an inherent seed dormancy. Shrub encroachment in alpine areas will likely depend on conditions that affect seed germination at the microsite-scale, despite overall conditions becoming more suitable for shrubs at high elevations.

## 1. Introduction

Across the arctic tundra and alpine areas of the world, woody shrub species are increasing in abundance [1,2]. This is partly due to the higher rate of climate warming and declines in snowpack that are occurring in these regions [3,4,5]. In general, woody shrubs require relatively warmer temperatures in order to maintain their carbon balance during periods of growth and reproduction [6,7] compared with herbaceous arctic and alpine plants. Hence, overall warming temperatures in many arctic and alpine regions are providing an opportunity for shrubs to increase their overall cover abundance, expand their range, and become dominant in parts of vegetation communities from which they were previously excluded, such as herbfields and grasslands [8,9,10,11]. While there is mounting evidence of shrub-cover change around the world, from repeat photography [8,12], long-term monitoring [13,14,15,16], field warming experiments [17,18] and dendrochronological studies [6,19], the mechanisms of shrub dispersal, recruitment and growth, and how well these operate with warming and variable temperatures, are less well understood.

The costs of seed production and risk of recruitment failure are high in cold-climate ecosystems where growing seasons are short and nutrients generally limited [20,21]. Thus, cold-climate regions are characterized by perennial, long-lived species with a dependence on clonal reproduction which emphasizes the adult life-history stages [22,23,24]. However, an increasing number of studies demonstrate that successful reproduction from seed is occurring in arctic and alpine areas and that these processes may be more common than previously thought [Vázquez-Ramírez and Venn 2021, (this issue) [25]. Recruitment via seed may also play an important role in the structure and functioning of alpine systems after disturbances such as fire, which may advantage obligate seeder species more than predominantly clonal species [26,27].

The encroachment of cold-climate shrub species into new regions occurs via three main processes: infilling by means of an increase in shrub cover through lateral growth of currently existing shrubs and recruitment between existing patches; an increase in growth including an increase in canopy height; or as an advancing shrubline or colonization of areas beyond the previous range limit [2]. Recruitment from seed and seedling survival underpins the third process. This has been evidenced by the continued encroachment of shrubs in these cold-climate regions over the last 50–100 years [2]. Regeneration via clonal means may play a lesser role in maintaining shrub populations and determining the future patterns of shrub dominance, particularly if warmer temperatures begin to tip the balance in favour of sexual recruitment in these regions.

The ‘germination niche’ describes the breadth of temperatures under which germination is possible. Germination is maximized at an optimum germination temperature, and then declines at lower and higher temperatures. The optimum shows some potential for acclimation and in theory may shift over time as species adapt to warmer conditions, but the shoulder temperatures, at the extremes of the germination niche, may become increasingly important for maintaining populations [28]. There is speculation that the breadth of the germination niche may be narrow for narrowly distributed species or those with very specific germination requirements, or broad for those that are widely distributed and/or have more general germination requirements [29,30]. As the climatic conditions in high mountain environments have increasingly become more favorable for shrub encroachment, generalist species occupying a wide geographic range and with a broad germination niche may have more opportunities for recruitment from seed. However, if such generalist shrub species are to succeed in these areas, they must also contend with the extreme temperatures present in high mountain areas, those that may have been preventing their encroachment until recent times. Thus, species with both a relatively high optimal temperature for germination and a broad germination niche will be advantaged under highly variable climatic conditions. Even if the optimum conditions for germination are not met, low germination rates at very cool or very warm temperatures may be sufficient to maintain populations in long-lived shrub species, providing that subsequent seedling survival and growth occurs.

Across the continental alpine areas of Australia (above treeline), there are approximately 40 woody shrub species from 13 families, some of which have a wide distribution that extends down from the alpine areas into sub-alpine and montane zones [31], while others are strict alpine endemics. Many alpine plant communities do have a shrub component; however, there are some communities such as frost-hollow grasslands and snowbeds where shrubs are generally excluded due to regular frost events and snow cover extending for long periods well into summer, respectively. In Australia, an increase in alpine shrub abundance matches a trend of rising temperatures spanning four decades and declining snowpack [8,9,32]. Yet, little is known about the mechanisms by which these shrub species are reproducing or specifically, the possibility of these species dispersing and recruiting from seed in environments where they were previously absent. In this study, we investigate the range of temperatures under which a selection of Australian alpine shrubs can germinate. We also attempt to relate the laboratory germination temperatures to long-term average temperature conditions at the location of the seed accession for each species, and across each species’ geographic range. Following this, we aim to understand whether species from more specialized habitats, with perhaps a narrower germination niche and fewer encroachment opportunities than a generalist shrub species might have, could germinate if suitable conditions arise. Specifically, we ask: (1) How broad is the germination niche of Australian alpine shrub species? (2) Does the maximum germination percentage for each species occur at a temperature within the range of temperatures expected at the seed accession site? (3) Does the germination niche breadth positively correlate with the temperature breadth across the species geographic distributions?

## 2. Materials and Methods

### 2.1. Australian Alpine Shrub Species Seed Sources

We chose 14 species representing eight families of high-elevation shrub species to include in this study (Table 1). Species were selected from the flora of woody species based on having a high elevation distribution throughout their range, being shrubs that are relatively common and representative of the vegetation, and there being seed available for research in conservation seed banks. Seed accession data including location and habitat descriptions were then used to select the highest elevation collection for each species, being careful to choose species from similar landscape positions and with similar community composition in the local habitat where possible (Appendix A).

### 2.2. Germination Trials

Two germination trials were performed over 2016 and 2017 in the Research School of Biology at the Australian National University, Canberra. The seeds from the Victorian Conservation Seed Bank were tested in 2016 and those from the Australian PlantBank were tested in 2017. Seed lots were assessed for viability prior to experimental treatments by squeeze and cut testing approximately 10 seeds of each lot. All species were exposed to cold stratification prior to germination to alleviate any physiological dormancy related to warm-cued germination [33] (*E. MacPhee pers. comm*.). Seeds were all cold stratified at 2 °C for 6 weeks in the dark on agar before entering the germination trials. The hard-coated seeds of *Hovea montana*, *Oxylobium ellipticum* and *Melicytus dentatus* were scarified with a scalpel to promote germination prior to cold stratification [34]. All seeds were plated in sterilized water agar in 32 mm petri dishes, with 10 or 20 seeds per dish, depending on seed size. For each germination trial, dishes were arranged in a stratified design across a gradient of constant temperatures between 5 and 45 °C using a Temperature Gradient Plate (TPG, Model GRD1, Grant Instruments, Cambridge, UK). Seven different temperatures were tested in 2016 and six different temperatures in 2017 (due to seed availability limitations) with three replicate petri dishes per species for each temperature (21 petri-dishes total in 2016 and 18 petri-dishes total in 2017 for each species) and with 12-hour days using fluorescent tube lights emitting up to 110 μmol m^−2^ s^−1^. Seeds were scored for germination, defined as radicle emergence, every 2–3 days for the first 6 weeks of each germination period, and then twice weekly followed by weekly as germination rates began to plateau around 6–9 weeks (depending on species). Remaining un-germinated seeds at the end of each trial were cut to determine viability and the final germination results were adjusted accordingly.

### 2.3. Data Analysis

In order to assess the association between laboratory germination niche breadth and germination conditions experienced in the field, we extracted multi-year average temperature conditions for each species at the seed accession location and across its geographic range. Specifically, we extracted data on two standard bioclimatic variables (maximum temperature the warmest month (BIO5); minimum temperature of the coldest month (BIO6)) from gridded datasets (1 km × 1 km grid resolution) in the New South Wales and Australian Capital Territory Regional Climate Modelling (NARCliM) project [35]. NARCLiM provides average historic climate conditions for a twenty-year baseline period (1990–2009) across south-east Australia generated from interpolation of meteorological station data. Geographic ranges were estimated from species occurrence data (latitude and longitude coordinates of collection locations) accessed from digitized herbarium records in the Australasian Virtual Herbarium (https://avh.chah.org.au/). Raw occurrence data were cleaned to remove taxonomic errors and spatial outliers prior to extraction of data on climate conditions. Coordinates of seed accession locations were provided by the seedbanks.

Using extracted data on climatic conditions, we calculated three temperature metrics for each species: (1) the Accession Temperature Range (AccRange), the difference between the minimum temperature of the coldest month and the maximum temperature of the warmest month at the accession site); (2) the Geographic Minimum Temperature Range (GeoMinRange), the lowest and highest minimum temperatures of the coldest month across the entire species geographic distribution; and (3) the Geographic Maximum Temperature Range (GeoMaxRange), the lowest and highest maximum temperatures of the warmest month across the entire species geographic distribution. All analyses were performed in R [36] using the *raster* package [37] in R.

## 3. Results

Of the 14 species chosen for this study, seven failed to germinate sufficiently to be analyzed further, with either no germination or only 1 seed germinating in total across the study. (Table 1). The highest mean germination proportions were recorded in the Asteraceae species *Ozothamnus alpinus* (83% at 35 °C) and *Olearia algida* (80% at 25 °C) (Figure 1). Germination results from the Ericaceae generally had very low mean germination proportions (0–19%), though this was not the case for one species—*Epacris paludosa* (55% at 25 °C). The lowest recorded mean germination proportion was recorded in *Epacris glacialis* with only 1.6% after accounting for viable seed; one seed germinating at 18 °C and one at 24 °C. More generally, the lowest temperature on the TPG in which any germination was recorded was in *Epacris paludosa* (11 °C and 11.5% mean germination) and *Olearia algida* (11 °C and 3% mean germination) (Figure 1). The breadth of the germination niche for all species tested ranged from 11–36 °C, with *Epacris paludosa* germinating across this entire range (Figure 1).

For each species, the seed accession temperature range (AccRange, the minimum temperature in the coldest month to the maximum temperature the warmest month at the specific seed accession site, Figure 1, green lines) was a poor indicator of the laboratory germination niche. In every case, germination occurred within the upper range of accession temperatures, and often 10–20 °C higher. The range of minimum and maximum temperatures across each species geographic range was also decoupled from the temperatures under which germination occurred on the TPG. No germination occurred for any species in the winter minimum temperature range (Figure 1, blue lines), while germination temperatures overlapped and often exceeded the range of summer high temperature across their geographic range (Figure 1, orange lines). For example, *Epacris petrophilla* was able to germinate at 5.6 °C higher than the maximum temperature across its geographic range (Figure 1e).

## 4. Discussion

Overall, the 14 shrub species selected for this study showed very low rates of germination. For most species, mean seed germination was less than 10% despite cold-stratification, scarification of the hard-coated seeds in the Fabaceae family, adequate time under experimental conditions and accounting for non-viable seeds with post-experiment cut-tests, of which there were very few. This suggests a deep inherent dormancy in many of these species, perhaps requiring multiple cycles of winter conditions, longer cold stratification, fire related cues (heat/smoke) [20], or possibly a longer time under suitable germination conditions to alleviate. This inherent dormancy potentially indicates that these seeds can persist in the soil seed bank under natural conditions for long periods of time and may only emerge in the standing vegetation once dormancy is alleviated and environmental conditions become suitable [38,39]. For the seven species in which we were able to alleviate dormancy, the seed germination results indicated an overall broad germination niche, spanning up to 17 °C for some species, despite very low germination in some species. Temperatures associated with the highest germination percentages were all above the range of temperatures present at each specific seed accession site according to long-term baseline climate records. While the breadth of the germination niche may have been reduced in some species by low germination rates, the optimum germination temperatures were consistently within or higher than the range of maximum temperatures modelled across the species’ geographic distribution. This finding may indicate opportunities for future germination under warmer ambient temperatures projected under climate change conditions, but will nonetheless require suitable microsite conditions as well, as we discuss below.

Our results demonstrate that for this suite of common alpine shrub species, germination is unlikely to be constrained by high temperatures. Further, ambient temperatures across species geographical ranges are unlikely to limit the regions where these species can germinate. Hence, as temperatures in alpine areas continue to rise [3], some alpine shrub species may continue with and possibly increase their reliance on germination across their range, undeterred by warming temperatures. In addition, any successful sexual recruitment via seed germination is likely to be coupled with improved clonal reproduction, as shrubs re-sprout above and below ground, and continue to build woody tissue in response to the warming temperatures [1].

Despite these potential opportunities for alpine shrub growth under a warming climate, there are several abiotic and biotic factors which may constrain shrub encroachment. For instance, the dispersal of seed into safe microsites for germination will likely remain a key limitation of shrub encroachment. Dispersal distances are likely to be constrained to under 10 m unless the seed has appendages to aide wind dispersal (e.g., such as a bristle pappus as is common in the Asteraceae) [40]. Some species have other specific germination requirements, including various moisture or fire-related cues such as heat and/or smoke derivatives [41].

Several of the study species, *Richea continentis*, *Pentachondra pumila*, *Melicytus dentatus* and *Tasmannia xerophila,* produce fleshy fruits and may require frugivores to promote seed germination via scarification of the seed coat or removal of germination inhibitors by separating the seeds from the pulp [42] and then aide in seed dispersal [43]. However, there is little evidence in the literature to suggest that functional frugivorous mutualisms exist in alpine Australia, nor are there any recognised frugivores in the region [44]. *Kunzea muelleri*, *Prostanthera cuneata* and Ericaceae flowers, seeds and occasionally leaves have reported to be eaten by *Burramys parvus* (mountain pygmy possum) [45], although, again, there is no documented evidence of a benefit for seed germination following defecation. Finally, some species may require microbial partners to facilitate seed germination [46] and there is evidence that some Ericaceae require a mycorrhizal partner to help seedlings establish [47]. Although, for the *Epacris* species investigated, there is a lack of specific knowledge about seed germination requirements and little evidence to suggest mycorrhizal associations are actually required for seed germination, rather they can germinate well with high germination rates once dormancy is overcome (*L. Guja pers. comm.*). The ability to make firm conclusions on this topic at this stage is therefore difficult and symbiotic drivers of germination in alpine species remains a substantial knowledge gap.

A large disconnect is apparent between the ambient air temperatures that are measured by the nationwide Bureau of Meteorology in a standard way (at 1.2 m and shaded inside a Stevenson Screen (http://bom.gov.au) and temperatures at scales relevant to germinating seeds—the microclimate. In alpine environments, extreme diurnal fluctuations in temperature often occur, and while these are likely to be extremely useful for promoting seed germination and can influence the success of emerging seedlings [48], they may not always be captured by interpolated climate data. For example, in mid-summer under sunny conditions in an alpine environment in south-eastern Australia, while air temperature at 10 cm above the ground surface was around 30 °C, leaf temperatures on the shrub *Prostanthera cuneata* were simultaneously 59 °C, and soil temperatures at 10 cm depth were around 19 °C (E. Sumner, *unpublished data*). The inferences we have made regarding the range of ambient temperatures across the shrub species’ geographic distribution were drawn from gridded datasets at a 1 km × 1 km resolution. In this respect, these estimated temperature ranges should only be used as a tentative source of generalized information about conditions in regions where germination might be possible. It may be the case that broad trends in germination niches associated with different climates and conditions are not commonly observed [30]. Field collected data on site specific climate conditions—although outside the scope of this work—would certainly provide more robust approximations of conditions at ground level within microsites where shrub seeds may be present. Therefore, the range of temperatures that we exposed shrub seeds to using the temperature gradient plate in the laboratory are likely to represent naturally occurring microsite temperatures, while the ambient temperatures interpolated across the species geographic distributions should be used with caution as they may be less relevant for germinating seeds.

Species with high proportions of germination, coupled with a broad range of possible germination temperatures, may indicate that they have additional options for recruitment and regeneration besides clonal reproduction. These species, in turn, could therefore become recognised as dominant encroaching shrub species. However, several research tasks remain for creating a stronger understanding of the associations between climate warming, seed germination requirements and shrub encroachment in alpine areas. Foremost, is determining the relative influence of regularly measured ambient temperatures and how these in turn affect the more relevant microsite scale, including soil moisture availability, on seed germination and seedling success. Repeated cycles of fire disturbance [26] and possible associations with symbionts will also need further attention to capture the real potential for alpine shrub germination. Despite a broad germination niche in a few shrub species in this study, overall, seed germination among alpine shrubs in south-eastern Australia may not be the most effective form of recruitment, with very low germination rates for some species. Consequently, the relative importance of clonal regeneration versus recruitment via seed needs a more thorough investigation [39,49] in order to make more accurate predictions of infilling of shrub cover between existing patches, simple increases in shrub growth and/or colonization of shrubs in areas beyond their current geographic distribution.

## Figures and Tables

**Figure 1 plants-10-00327-f001:**
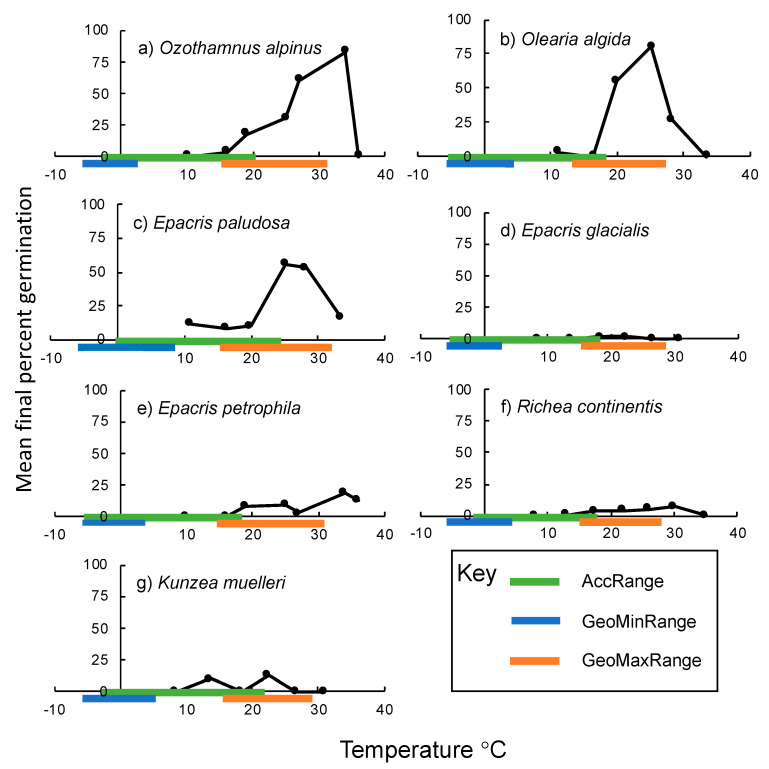
The germination niche as described by the mean final percent germination across a gradient of temperatures for the seven study species (**a**) *Ozothamnus alpinus,* (**b**) *Olearia algida*, (**c**) *Epacris paludosa*, (**d**) *Epacris glacialis*, (**e**) *Epacris petrophila,* (**f**) *Richea continentis*, (**g**) *Kunzea muelleri*. Temperature metrics for each seed accession site and species geographic range, as extracted from gridded average historic climate datasets (1990–2009), are indicated by colored lines: Green lines (AccRange) indicate the seed accession temperature range, the minimum temperature in the coldest month to the maximum temperature the warmest month at the accession site; blue lines (GeoMinRange) indicate the lowest and highest minimum temperatures of the coldest month across the entire species geographic distribution; and orange lines (GeoMaxRange) indicate the lowest and highest maximum temperatures of the warmest month across the entire species geographic distribution.

**Table 1 plants-10-00327-t001:** The families, species and their seed bank source used in this study. Species marked with an asterisk* failed to germinate sufficiently to be included in further data analysis. VCSB denotes Victorian Conservation Seed Bank, Royal Botanic Gardens Victoria, APB denotes Australian PlantBank, Australian Botanic Garden Mount Annan.

Family	Species	Seed Bank Source
Asteraceae	*Ozothamnus alpinus*	VCSB
	*Olearia algida*	APB
Ericaceae	*Epacris glacialis*	APB
	*Epacris paludosa*	APB
	*Epacris petrophila*	VCSB
	*Pentachondra pumila **	VCSB
	*Richea continentis*	VCSB
Fabaceae	*Hovea montana **	VCSB
	*Oxylobium ellipticum **	VCSB
Lamiaceae	*Prostanthera cuneata **	VCSB
Myrtaceae	*Kunzea muelleri*	APB
Thymaleaceae	*Pimelea ligustrina **	APB
Violaceae	*Melicytus dentatus **	APB
Winteraceae	*Tasmannia xerophila **	VCSB

## Data Availability

The data for this study are available by contacting R.V.G. at rachael.gallagher@mq.edu.au.

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
