# Peer review of "Germination at Extreme Temperatures: Implications for Alpine Shrub Encroachment"

_plants, 2021, doi:10.3390/plants10020327_

Round 1

Reviewer 1 Report

Unfortunately, I cannot recommend this work for publication, as its level is clearly below the high standards set by Plants magazine. The work practically does not carry new data, does not reveal new mechanisms or factors of plant life.

The authors ignored the most interesting point with non-germinated seeds. They just excluded these species from the analysis! Why didn't they sprout? Maybe the conditions of stratification were not suitable? Or the seeds do not have enough time, maybe their embryo is underdeveloped and they need several years to germinate (for example, like a peony)? The authors do not address this issue. Only in the discussion do they put forward hypotheses based on the literature, but this is speculative!

The authors only showed that the optimum germination of seeds of their studied species (and even then not all, but which they were able to germinate) is shifted relative to the conditions in which they grow, but is within the boundaries of a certain physiological norm. The reasonably warmer, the better the seeds germinate. This is absolutely clear! The whole discussion about the possible distribution of species is not consistent, since no research has been done in this area. The authors did not show any mechanism or even an intracellular factor promoting seed germination. The work is methodically very poor and more like a school experience.

Author Response

Thanks for reading our manuscript. We have now included some additional reasons as to why some of the seeds may not have germinated in the Discussion, as per your suggestions.  

Reviewer 2 Report

The ms is well writen and it was nice to read.

My main consern is the low number of replications in the experiment.

I know, this is a problem of unsufficient material.

May be the problem can be solved.

Author Response

Thank you for reading our manuscript and providing some constructive criticism that we can address. We have now improved the manuscript in the following ways according to your comments:

  • 6 weeks of cold stratification is a standard length of time for Australian alpine species - some low alpine and sub-alpine places would only get about  6 weeks of snow cover. Many of the seeds were collected from such places.  Also, 2C is also a suitable temperature, as soils rarely freeze (unless they are exposed) in the mountains.    
  • The issue of adequate replicate petri-dishes per temperature (only 3) was difficult to improve due to seed availability and space limitations on the temperature gradient plate. However, we have made the text regarding the total number of petri-dishes clearer - 21 total dishes of each species in 2017 and 18 total dishes of each species in 2016. 
  • We cited Ladinig et al. (2015) to help describe the high and variable temperatures of different plant parts, instead of Reisigl and Keller (1987) which we could not access.
  • We have revised the supplementary table to be clearer (we submitted this  in a landscape format that was changed by the journal) and included seed collection dates 

Reviewer 3 Report

The title looks well.
Abstract is generally good. However, it could be improved by making it more informative (e.g. adding names of all analysed shrub species; adding the study period). Key words seem to be alright, but I would recommend to expand this list until 7-8 key words.

The section Introduction is well supported by references and it provides a good overview for background of this study. Nevertheless, the section contains irrelevant and superfluous material. For example, there is no reason to list the classification of Myers-Smith et al. (2011) in extensive details. Moreover, I recommend to present the all approaches and classifications used in the study in the section Materials and Methods.
Additionally (lines 88-92), the section Introduction should not include statement(s) about what did authors study during the research period. This information can be presented in the Material and Methods (as a statement, what methods and approaches were used in the study) or in the section Results (as statements of the results obtained by authors).

The section Material and Methods needs some corrections. For example, its first paragraph (lines 101-107) is rather appropriate to the section Introduction
Sentence in lines 116-118 should be re-written because now it presents results (what did authors obtain).
In line 157, "was" should be changed on "were" (data... were).

The section Results looks good and appropriate. Just some corrections are needed here:
I recommend to change "percentages" on "proportions", e.g., in lines 172, 175 and others.
Please, correct the use of percent signs (with or without space between it and number).
In the first paragraph (lines 170-182), authors pay attention to the highest and lowest germination proportions, while I would suggest to mention also the germination rates under temperature conditions in sites where the seeds were harvested (i.e. mean AccRange, I guess).
Then, Fig.1 needs in minor revision. So, now the coloured (AccRange, GeoMinRange, GeoMaxRange) lines were obviously drawn handily. I assumed this because these lines are not drawn neatly enough as if they would be drawn using Paint tool. I suggest to re-make this Figure by correcting these coloured lines to be more accurate. In addition, please, add only titles of temperature designations (i.e. AccRange, GeoMinRange, GeoMaxRange), because definitions of these temperatures are already mentioned in the figure caption.

The section Discussion is relatively appropriate. But I have some minor suggestions as follows:
In line 229, please, support the statement "as temperatures in alpine areas continue to rise" by reference(s).
In line 226, please, delete the superfluous comma after "that".
In line 235, I suggest change "climates" on "climate changes". However, I would suggest to be more accurate by discussing exactly temperature increase, but not climate changes in general, because in this paper, authors studied only this climatic parameter.

Finally, I think that this study is a well written paper with a need in minor corrections. However, I would suggest trying to assume (based on the obtained data on seed germination of shrubs), which of the studied shrubs could become dominants under further increase in annual temperature of the studied area.

Author Response

Thank you for reading our manuscript and providing some comments and suggestions for improvement. We address the main points below:

  • In terms of Abstract length - we are already at the 200 word maximum limit and are unable to add any more words
  • We have added some more key words which are not already in the title
  • The classification of shrub expansion as per Myers-Smith et al. (2011) has been re-phrased in order to inform the reader, as well as keep the paragraph's sentiment intact.
  • We have removed the reference to the 'Temperature Gradient Plate' towards the end of the Introduction, and agree that the use of this instrument is adequately described in the Methods section. However, we have kept the rest of the sentence as it helps to put the aims of the study into context.
  • The first paragraph of the Methods has indeed been moved to the Introduction as per your suggestion.
  • The sentence in lines 116-118 describing results has been removed.
  • 'was' changed to 'were' as per your suggestion 
  • The use of % signs with no spaces between the number and symbol has been fixed, The word 'proportions' has been used instead of 'percentage' in most instances where appropriate.
  • We have chosen not to elaborate about the range of temperatures under which germination occurred for all species in the Results text, and instead suggest that referring the reader to Figure 1 which shows the range of these temperatures is sufficient.
  • The AccRange, GeoMinRange and GeoMaxRange coloured lines have been fixed and look clear. We have simplified the labels in the Key and updated the Figure caption    
  • We addressed the grammatical points raised in the Discussion, and included a sentence about which species may become dominant encroaching shrubs.  

Round 2

Reviewer 1 Report

Dear authors, unfortunately I do not see any significant changes in your article so that now I can recommend it for publication.